# THIEVES ON SESAME STREET!
# MODEL EXTRACTION OF BERT-BASED APIS

**Kalpesh Krishna**[*]
CICS, UMass Amherst
kalpesh@cs.umass.edu

**Gaurav Singh Tomar**
Google Research
gtomar@google.com

**Ankur P. Parikh**
Google Research
aparikh@google.com

**Nicolas Papernot**
Google Research
papernot@google.com

**Mohit Iyyer**
CICS, UMass Amherst
miyyer@cs.umass.edu

## ABSTRACT

We study the problem of model extraction in natural language processing, in which an adversary with only query access to a victim model attempts to reconstruct a local copy of that model. Assuming that both the adversary and victim model fine-tune a large pretrained language model such as BERT (Devlin et al., 2019), we show that the adversary does not need any real training data to successfully mount the attack. In fact, the attacker need not even use grammatical or semantically meaningful queries: we show that *random* sequences of words coupled with task-specific heuristics form effective queries for model extraction on a diverse set of NLP tasks, including natural language inference and question answering. Our work thus highlights an exploit only made feasible by the shift towards transfer learning methods within the NLP community: for a query budget of a few hundred dollars, an attacker can extract a model that performs only slightly worse than the victim model. Finally, we study two defense strategies against model extraction—membership classification and API watermarking—which while successful against naive adversaries, are ineffective against more sophisticated ones.

## 1 INTRODUCTION

Machine learning models represent valuable intellectual property: the process of gathering training data, iterating over model design, and tuning hyperparameters costs considerable money and effort. As such, these models are often only indirectly accessible through web APIs that allow users to query a model but not inspect its parameters. Malicious users might try to sidestep the expensive model development cycle by instead locally reproducing an existing model served by such an API. In these attacks, known as "model stealing" or "model extraction" (Lowd & Meek, 2005; Tramèr et al., 2016), the adversary issues a large number of queries and uses the collected (input, output) pairs to train a local copy of the model. Besides theft of intellectual property, extracted models may leak sensitive information about the training data (Tramèr et al., 2016) or be used to generate adversarial examples that evade the model served by the API (Papernot et al., 2017).

With the recent success of contextualized pretrained representations for transfer learning, NLP models created by finetuning ELMo (Peters et al., 2018) and BERT (Devlin et al., 2019) have become increasingly popular (Gardner et al., 2018). Contextualized pretrained representations boost performance and reduce sample complexity (Yogatama et al., 2019), and typically require only a shallow task-specific network—sometimes just a single layer as in BERT. While these properties are advantageous for representation learning, we hypothesize that they also make model extraction easier.

In this paper,[1] we demonstrate that NLP models obtained by fine-tuning a pretrained BERT model can be extracted even if the adversary does not have access to *any* training data used by the API

---

[*]Work done during an internship at Google Research.

[1]All the code necessary to reproduce experiments in this paper can be found in https://github.com/google-research/language/tree/master/language/bert_extraction.

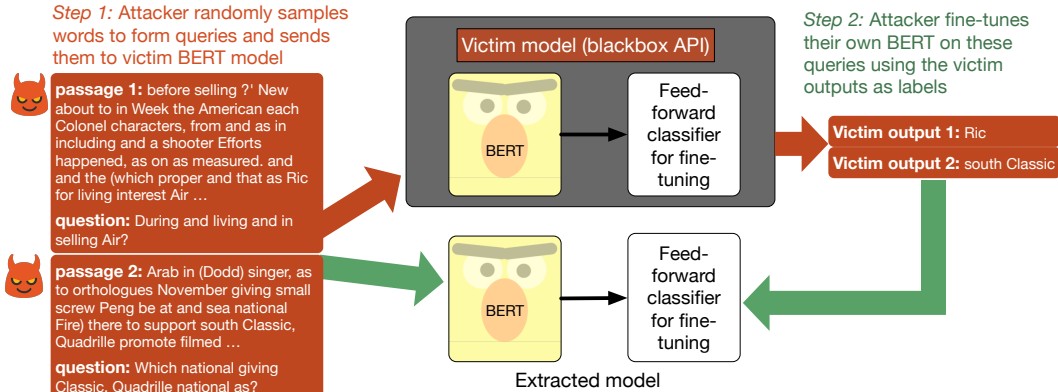

Figure 1: Overview of our model extraction setup for question answering.[2] An attacker first queries a victim BERT model, and then uses its predicted answers to fine-tune their own BERT model. This process works even when passages and questions are random sequences of words as shown here.

provider. In fact, the adversary does not even need to issue well-formed queries: our experiments show that extraction attacks are possible even with queries consisting of *randomly sampled sequences of words* coupled with simple task-specific heuristics (Section 3). While extraction performance improves further by leveraging sentences and paragraphs from Wikipedia (Section 4), the fact that random word sequences are sufficient to extract models contrasts with prior work, where large-scale attacks require at minimum that the adversary can access a small amount of semantically-coherent data relevant to the task (Papernot et al., 2017; Correia-Silva et al., 2018; Orekondy et al., 2019a; Pal et al., 2019; Jagielski et al., 2019). These attacks are cheap: our most expensive attack cost around $500, estimated using rates of current API providers.

In Section 5.1, we perform a fine-grained analysis of the randomly-generated queries. Human studies on the random queries show that despite their effectiveness in extracting good models, they are mostly nonsensical and uninterpretable, although queries closer to the original data distribution work better for extraction. Furthermore, we discover that pretraining on the attacker's side makes model extraction easier (Section 5.2).

Finally, we study the efficacy of two simple defenses against extraction — membership classification (Section 6.1) and API watermarking (Section 6.2) — and find that while they work well against naive adversaries, they fail against adversaries who adapt to the defense. We hope that our work spurs future research into stronger defenses against model extraction and, more generally, on developing a better understanding of why these models and datasets are particularly vulnerable to such attacks.

## 2 RELATED WORK

We relate our work to prior efforts on model extraction, most of which have focused on computer vision applications. Because of the way in which we synthesize queries for extracting models, our work also directly relates to zero-shot distillation and studies of rubbish inputs to NLP systems.

**Model extraction attacks** have been studied both empirically (Tramèr et al., 2016; Orekondy et al., 2019a; Juuti et al., 2019) and theoretically (Chandrasekaran et al., 2018; Milli et al., 2019), mostly against image classification APIs. These works generally synthesize queries in an active learning setup by searching for inputs that lie close to the victim classifier's decision boundaries. This method does not transfer to text-based systems due to the discrete nature of the input space.[3] The only prior work attempting extraction on NLP systems is Pal et al. (2019), who adopt pool-based active learning to select natural sentences from WikiText-2 and extract 1-layer CNNs for tasks expecting

---

[2]The BERT clipart in this figure was originally used in `http://jalammar.github.io/illustrated-bert/`.

[3]In our initial experiments we tried equivalent active learning algorithms with the HotFlip algorithm (Ebrahimi et al., 2018) but had limited success.

| Task | RANDOM example | WIKI example |
|------|----------------|--------------|
| SST2 | cent 1977, preparation (120 remote Program finance add broader protection ( **76.54**% negative) | So many were produced that thousands were Brown's by coin 1973 (**98.59**% positive) |
| MNLI | **P**: Mike zone fights Woods Second State known, defined come
**H**: Mike zone released, Woods Second HMS males defined come (**99.89**% contradiction) | **P**: voyage have used a variety of methods to Industrial their Trade
**H**: descent have used a officially of methods exhibition Industrial their Trade (**99.90**% entailment) |
| SQuAD | **P**: a of Wood, curate him and the " Stop Alumni terrestrial the of of roads Kashyap. Space study with the Liverpool, Wii Jordan night Sarah lbf a Los the Australian three English who have that that health officers many new workforce...
**Q**: *How workforce. Stop who new of Jordan et Wood, displayed the?*
**A**: **Alumni terrestrial the of of roads Kashyap** | **P**: Since its release, Dookie has been featured heavily in various "must have" lists compiled by the music media. Some of the more prominent of these lists to feature Dookie are shown below; this information is adapted from Acclaimed Music.
**Q**: *What are lists feature prominent " adapted Acclaimed are various information media.?*
**A**: **"must have"** |

Table 1: Representative examples from the extraction datasets, highlighting the effect of task-specific heuristics in MNLI and SQuAD. More examples in Appendix A.5.

single inputs. In contrast, we study a more realistic extraction setting with *nonsensical* inputs on *modern BERT-large* models for tasks expecting *pairwise* inputs like question answering.

Our work is related to prior work on **data-efficient distillation**, which attempts to distill knowledge from a larger model to a small model with access to limited input data (Li et al., 2018) or in a zero-shot setting (Micaelli & Storkey, 2019; Nayak et al., 2019). However, unlike the model extraction setting, these methods assume white-box access to the teacher model to generate data impressions.

**Rubbish inputs**, which are randomly-generated examples that yield high-confidence predictions, have received some attention in the model extraction literature. Prior work (Tramèr et al., 2016) reports successful extraction on SVMs and 1-layer networks using i.i.d noise, but no prior work has scaled this idea to deeper neural networks for which a single class tends to dominate model predictions on most noise inputs (Micaelli & Storkey, 2019; Pal et al., 2019). Unnatural text inputs have previously been shown to produce overly confident model predictions (Feng et al., 2018), break translation systems (Belinkov & Bisk, 2018), and trigger disturbing outputs from text generators (Wallace et al., 2019). In contrast, here we show their effectiveness at *training* models that work well on real NLP tasks *despite not seeing any real examples* during training.

## 3 METHODOLOGY

**What is BERT?** We study model extraction on BERT, **B**idirectional **E**ncoder **R**epresentations from **T**ransformers (Devlin et al., 2019). BERT-large is a 24-layer transformer (Vaswani et al., 2017), $f_{\text{bert},\theta}$, which converts a word sequence $x = (x^1, ..., x^n)$ of length $n$ into a high-quality sequence of vector representations $\mathbf{v} = (\mathbf{v}^1, ..., \mathbf{v}^n)$. These representations are contextualized — every vector $\mathbf{v}^i$ is conditioned on the whole sequence $x$. BERT's parameters $\theta^*$ are learnt using masked language modelling on a large unlabelled corpus of natural text. The public release of $f_{\text{bert},\theta^*}$ revolutionized NLP, as it achieved state-of-the-art performance on a wide variety of NLP tasks with minimal task-specific supervision. A modern NLP system for task $T$ typically leverages the fine-tuning methodology in the public BERT repository:[4] a task-specific network $f_{T,\phi}$ (generally, a 1-layer feedforward network) with parameters $\phi$ expecting $\mathbf{v}$ as input is used to construct a composite function $g_T = f_{T,\phi} \circ f_{\text{bert},\theta}$. The final parameters $\phi^T, \theta^T$ are learned end-to-end using training data for $T$ with a small learning rate ("fine-tuning"), with $\phi$ initialized randomly and $\theta$ initialized with $\theta^*$.

**Description of extraction attacks:** Assume $g_T$ (the "victim model") is a commercially available black-box API for task $T$. A malicious user with black-box query access to $g_T$ attempts to reconstruct a local copy $g'_T$ (the "extracted model"). Since the attacker does not have training data for $T$, they use a task-specific query generator to construct several possibly nonsensical word sequences $\{x_i\}_1^m$ as queries to the victim model. The resulting dataset $\{x_i, g_T(x_i)\}_1^m$ is used to train $g'_T$.

---

[4]https://github.com/google-research/bert

| Task | # Queries | Cost | Model | Accuracy | Agreement |
|------|-----------|------|-------|----------|-----------|
| SST2 | 67349 | $62.35 | VICTIM | 93.1% | - |
| | | | RANDOM | 90.1% | 92.8% |
| | | | WIKI | 91.4% | 94.9% |
| | | | WIKI-ARGMAX | 91.3% | 94.2% |
| MNLI | 392702 | $387.82* | VICTIM | 85.8% | - |
| | | | RANDOM | 76.3% | 80.4% |
| | | | WIKI | 77.8% | 82.2% |
| | | | WIKI-ARGMAX | 77.1% | 80.9% |
| SQuAD 1.1 | 87599 | $115.01* | VICTIM | 90.6 F1, 83.9 EM | - |
| | | | RANDOM | 79.1 F1, 68.5 EM | 78.1 F1, 66.3 EM |
| | | | WIKI | 86.1 F1, 77.1 EM | 86.6 F1, 77.6 EM |
| BoolQ | 9427 | $5.42* | VICTIM | 76.1% | - |
| | | | WIKI | 66.8% | 72.5% |
| | | | WIKI-ARGMAX | 66.0% | 73.0% |
| | 471350 | $516.05* | WIKI (50x data) | 72.7% | 84.7% |

Table 2: A comparison of the original API (VICTIM) with extracted models (RANDOM and WIKI) in terms of **Accuracy** on the original development set and **Agreement** between the extracted and victim model on the development set inputs. Notice high accuracies for extracted models. Unless specified, all extraction attacks were conducted use the same number of queries as the original training dataset. The * marked costs are estimates from available Google APIs (details in Appendix A.2).

Specifically, we assume that the attacker fine-tunes the public release of $f_{\text{bert}, \theta^*}$ on this dataset to obtain $g'_T$.[5] A schematic of our extraction attacks is shown in Figure 1.

**NLP tasks:** We extract models on four diverse NLP tasks that have different kinds of input and output spaces: (1) binary sentiment classification using SST2 (Socher et al., 2013), where the input is a single sentence and the output is a probability distribution between *positive* and *negative*; (2) ternary natural language inference (NLI) classification using MNLI (Williams et al., 2018), where the input is a pair of sentences and the output is a distribution between *entailment*, *contradiction* and *neutral*; (3) extractive question answering (QA) using SQuAD 1.1 (Rajpurkar et al., 2016), where the input is a paragraph and question and the output is an answer span from the paragraph; and (4) boolean question answering using BoolQ (Clark et al., 2019), where the input is a paragraph and question and the output is a distribution between *yes* and *no*.

**Query generators:** We study two kinds of query generators, RANDOM and WIKI. In the RANDOM generator, an input query is a nonsensical sequence of words constructed by sampling[6] a Wikipedia vocabulary built from WikiText-103 (Merity et al., 2017). In the WIKI setting, input queries are formed from actual sentences or paragraphs from the WikiText-103 corpus. We found these two generators insufficient by themselves to extract models for tasks featuring complex interactions between different parts of the input space (e.g., between premise and hypothesis in MNLI or question and paragraph in SQuAD). Hence, we additionally apply the following task-specific heuristics:

- MNLI: since the premise and hypothesis often share many words, we randomly replace three words in the premise with three random words to construct the hypothesis.

- SQuAD / BoolQ: since questions often contain words in the associated passage, we uniformly sample words from the passage to form a question. We additionally prepend a question starter word (like "*what*") to the question and append a ? symbol to the end.

Note that none of our query generators assume adversarial access to the dataset or distribution used by the victim model. For more details on the query generation, see Appendix A.3. Representative example queries and their outputs are shown in Table 1. More examples are provided in Appendix A.5.

---

[5]We experiment with alternate attacker networks in Section 5.2.

[6]We use uniform random sampling for SST2 / MNLI and unigram frequency-based sampling for SQuAD / BoolQ. Empirically, we found this setup to be the most effective in model extraction.

| Task | Model | 0.1x | 0.2x | 0.5x | 1x | 2x | 5x | 10x |
|------|-------|------|------|------|-----|-----|-----|------|
| SST2 | VICTIM | 90.4 | 92.1 | 92.5 | 93.1 | - | - | - |
|      | RANDOM | 75.9 | 87.5 | 89.0 | 90.1 | 90.5 | 90.4 | 90.1 |
| (**1x** = 67349) | WIKI | 89.6 | 90.6 | 91.7 | 91.4 | 91.6 | 91.2 | 91.4 |
| MNLI | VICTIM | 81.9 | 83.1 | 85.1 | 85.8 | - | - | - |
|      | RANDOM | 59.1 | 70.6 | 75.7 | 76.3 | 77.5 | 78.5 | 77.6 |
| (**1x** = 392702) | WIKI | 68.0 | 71.6 | 75.9 | 77.8 | 78.9 | 79.7 | 79.3 |
| SQuAD 1.1 | VICTIM | 84.1 | 86.6 | 89.0 | 90.6 | - | - | - |
|           | RANDOM | 60.6 | 68.5 | 75.8 | 79.1 | 81.9 | 84.8 | 85.8 |
| (**1x** = 87599) | WIKI | 72.4 | 79.6 | 83.8 | 86.1 | 87.4 | 88.4 | 89.4 |
| BoolQ | VICTIM | 63.3 | 64.6 | 69.9 | 76.1 | - | - | - |
| (**1x** = 9427) | WIKI | 62.1 | 63.1 | 64.7 | 66.8 | 67.6 | 69.8 | 70.3 |

Table 3: Development set accuracy of various extracted models on the original development set at different query budgets expressed as fractions of the original dataset size. Note the high accuracies for some tasks even at low query budgets, and diminishing accuracy gains at higher budgets.

# 4 EXPERIMENTAL VALIDATION OF OUR MODEL EXTRACTION ATTACKS

First, we evaluate our extraction procedure in a controlled setting where an attacker uses an identical number of queries as the original training dataset (Table 2); afterwards, we investigate different query budgets for each task (Table 3). We provide commercial cost estimates for these query budgets using the Google Cloud Platform's Natural Language API calculator.[7] We use two metrics for evaluation: *Accuracy* of the extracted models on the original development set, and *Agreement* between the outputs of the extracted model and the victim model on the original development set inputs. Note that these metrics are defined at a label level — metrics are calculated using the argmax labels of the probability vectors predicted by the victim and extracted model.

In our controlled setting (Table 2), our extracted models are surprisingly accurate on the original development sets of all tasks, even when trained with nonsensical inputs (RANDOM) that do not match the original data distribution.[8] Accuracy improves further on WIKI: *extracted SQuAD models recover 95% of original accuracy despite seeing only nonsensical questions during training*. While extracted models have high accuracy, their agreement is only slightly better than accuracy in most cases. Agreement is even lower on held-out sets constructed using the WIKI and RANDOM sampling scheme. On SQuAD, extracted WIKI and RANDOM have low agreements of 59.2 F1 and 50.5 F1 *despite being trained on identically distributed data*. This indicates poor functional equivalence between the victim and extracted model as also found by Jagielski et al. (2019). An ablation study with alternative query generation heuristics for SQuAD and MNLI is conducted in Appendix A.4.

**Classification with argmax labels only:** For classification datasets, we assumed the API returns a probability distribution over output classes. This information may not be available to the adversary in practice. To measure what happens when the API only provides argmax outputs, we re-run our WIKI experiments for SST2, MNLI and BoolQ with argmax labels and present our results in Table 2 (WIKI-ARGMAX). We notice a minimal drop in accuracy from the corresponding WIKI experiments, indicating that access to the output probability distribution is not crucial for model extraction. Hence, hiding the full probability distribution is not a viable defense strategy.

**Query efficiency:** We measure the effectiveness of our extraction algorithms with varying query budgets, each a different fraction of the original dataset size, in Table 3. Even with small query

---

[7]The calculator can be found in `https://cloud.google.com/products/calculator/`. Since Google Cloud's API does not provide NLI and QA models, we base our estimates off the costs of the Entity Analysis and Sentiment Analysis APIs. All costs calculated on the original datasets by counting every 1000 characters of the input as a different unit. More details on pricing in Appendix A.2.

[8]We omit BoolQ / RANDOM from the table as it failed to converge, possibly due to either the sparse signal from *yes* / *no* outputs for a relatively complex classification task, or the poor accuracy of the victim model which reduces extraction signal. The victim model achieves just 76.1% binary accuracy compared to the majority class of 62.1%.

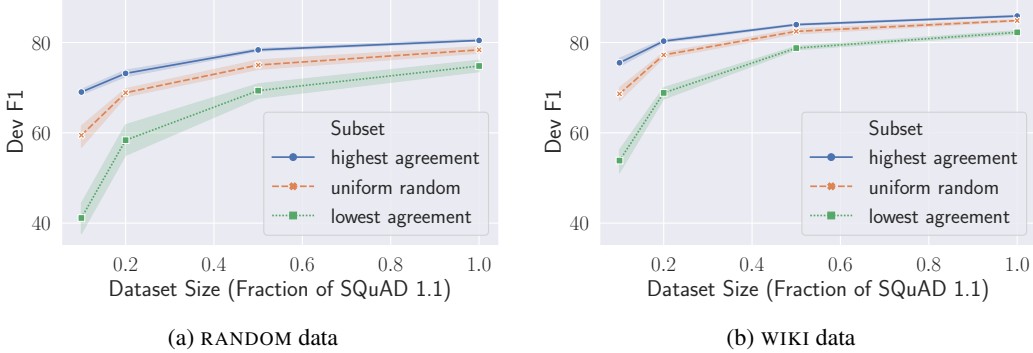

(a) RANDOM data          (b) WIKI data

Figure 2: Average dev F1 for extracted SQuAD models after selecting different subsets of data from a large pool of WIKI and RANDOM data. Subsets are selected based on the agreement between the outputs of different runs of the original SQuAD model. Notice the large difference between the highest agreement (blue) and the lowest agreement (green), especially at small dataset sizes.

budgets, extraction is often successful; while more queries is usually better, accuracy gains quickly diminish. Approximate costs for these attacks can be extrapolated from Table 2.

## 5 ANALYSIS

These results bring many natural questions to mind. What properties of nonsensical input queries make them so amenable to the model extraction process? How well does extraction work for these tasks without using large pretrained language models? In this section, we perform an analysis to answer these questions.

### 5.1 A CLOSER LOOK AT NONSENSICAL QUERIES

Previously, we observed that nonsensical input queries are surprisingly effective for extracting NLP models based on BERT. Here, we dig into the properties of these queries in an attempt to understand why models trained on them perform so well. Do different victim models produce the same answer when given a nonsensical query? Are some of these queries better for extraction? Did our task-specific heuristics perhaps make these nonsensical queries "interpretable" to humans in some way? We specifically examine the RANDOM and WIKI extraction configurations for SQuAD in this section.

**Do different victim models agree on the answers to nonsensical queries?** We train five victim SQuAD models on the original training data with identical hyperparameters, varying only the random seed; each achieves an F1 between 90 and 90.5. Then, we measure the average pairwise F1 ("agreement") between the answers produced by these models for different types of queries. As expected, the models agree very frequently when queries come from the SQuAD training set (96.9 F1) or development set (90.4 F1). However, their agreement drops significantly on WIKI queries (53.0 F1) and even further on RANDOM queries (41.2 F1).[9] Note that this result parallels prior work (Lakshminarayanan et al., 2017), where an ensemble of classifiers has been shown to provide better uncertainty estimates and out-of-distribution detection than a single overconfident classifier.

**Are high-agreement queries better for model extraction?** While these results indicate that on average, victim models tend to be brittle on nonsensical inputs, it is possible that high-agreement queries are more useful than others for model extraction. To measure this, we sort queries from our 10x RANDOM and WIKI datasets according to their agreement and choose the highest and lowest agreement subsets, where subset size is a varying fraction of the original training data size (Figure 2). We observe large F1 improvements when extracting models using high-agreement subsets, consistently beating random and low-agreement subsets of identical sizes. This result shows that agreement between victim models is a good proxy for the quality of an input-output pair for extrac-

---

[9]We plot a histogram of the agreement in Appendix A.1.

tion. Measuring this agreement in extracted models and integrating this observation into an active learning objective for better extraction is an interesting direction that we leave to future work.

**Are high-agreement nonsensical queries interpretable to humans?** Prior work (Xu et al., 2016; Ilyas et al., 2019) has shown deep neural networks can leverage non-robust, uninterpretable features to learn classifiers. Our nonsensical queries are not completely random, as we do apply task-specific heuristics. Perhaps as a result of these heuristics, do high-agreement nonsensical textual inputs have a human interpretation? To investigate, we asked three human annotators[10] to answer twenty SQuAD questions from each of the WIKI and RANDOM subsets that had unanimous agreement among victim models, and twenty original SQuAD questions as a control. On the WIKI subset, annotators matched the victim models' answer exactly 23% of the time (33 F1). Similarly, a 22% exact match (32 F1) was observed on RANDOM. In contrast, annotators scored significantly higher on original SQuAD questions (77% exact match, 85 F1 against original answers). Interviews with the annotators revealed a common trend: annotators used a word overlap heuristic (between the question and paragraph) to select entities as answer spans. While this heuristic partially interprets the extraction data's signal, most of the nonsensical question-answer pairs remain mysterious to humans. More details on inter-annotator agreement are provided in Appendix A.6.

## 5.2 THE IMPORTANCE OF PRETRAINING

So far we assumed that the victim and the attacker both fine-tune a pretrained BERT-large model. However, in practical scenarios, the attacker might not have information about the victim architecture. What happens when the attacker fine-tunes a different base model than the victim? What if the attacker extracts a QA model from scratch instead of fine-tuning a large pretrained language model? Here, we examine how much the extraction accuracy depends on the pretraining setup.

**Mismatched architectures:** BERT comes in two different sizes: the 24 layer BERT-large and the 12 layer BERT-base. In Table 4, we measure the development set accuracy on MNLI and SQuAD when the victim and attacker use different configurations of these two models. We notice that accuracy is always higher when the attacker starts from BERT-large, *even when the victim was initialized with BERT-base*. Additionally, given a fixed attacker architecture, accuracy is better when the victim uses the same model (e.g., if the attacker starts from BERT-base, they will have better results if the victim also used BERT-base).

| Victim | Attacker | MNLI | SQuAD (WIKI) |
|---|---|---|---|
| BERT-large | BERT-large | 77.8% | 86.1 F1, 77.1 EM |
| BERT-base | BERT-large | 76.3% | 84.2 F1, 74.8 EM |
| BERT-base | BERT-base | 75.7% | 83.0 F1, 73.4 EM |
| BERT-large | BERT-base | 72.5% | 81.2 F1, 71.3 EM |

Table 4: Development set accuracy using WIKI queries on MNLI and SQuAD with mismatched BERT architectures between the victim and attacker. Note the trend: (large, large) > (base, large) > (base, base) > (large, base) where the (·, ·) refers to (victim, attacker) pretraining.

Next, we experiment with an alternative non-BERT pretrained language model as the attacker architecture. We use XLNet-large (Yang et al., 2019), which has been shown to outperform BERT-large in a large variety of downstream NLP tasks. In Table 5, we compare XLNet-large and BERT-large attacker architectures keeping a fixed BERT-large victim architecture. Note the superior performance of XLNet-large attacker models on SQuAD compared to BERT-large in both RANDOM and WIKI attack settings, *despite seeing a mismatched victim's (BERT-large) outputs during training*.

Our experiments are reminiscent of similar discussion in Tramèr et al. (2016) on *Occam Learning*, or appropriate alignment of victim-attacker architectures. Overall, the results suggest that attackers can maximize their accuracy by fine-tuning more powerful language models, and that matching architectures is a secondary concern.

---

[10]Annotators were English-speaking graduate students who voluntarily agreed to participate and were completely unfamiliar with our research goals.

| Attacker | Training Data X | Training Data Y | SQuAD |
|---|---|---|---|
| BERT-large | ORIGINAL X | ORIGINAL Y | 90.6 F1 |
| XLNet-large | ORIGINAL X | ORIGINAL Y | **92.8** F1 |
| BERT-large | RANDOM X | BERT-LARGE Y | 86.1 F1 |
| XLNet-large | RANDOM X | BERT-LARGE Y | **89.2** F1 |
| BERT-large | WIKI X | BERT-LARGE Y | 79.1 F1 |
| XLNet-large | WIKI X | BERT-LARGE Y | **80.9** F1 |

Table 5: SQuAD dev set results comparing BERT-large and XLNet-large attacker architectures. Note the effectiveness of XLNet-large over BERT-large in both RANDOM and WIKI attack settings, despite seeing BERT-LARGE victim outputs during training. *Legend*: **Training Data X, Y** represent the input and output pairs used while training the attacker model; ORIGINAL represents the original SQuAD dataset; BERT-LARGE represents the outputs from the victim BERT-large model.

**What if we train from scratch?** Fine-tuning BERT or XLNet seems to give attackers a significant headstart, as only the final layer of the model is randomly initialized and the BERT parameters start from a good initialization representative of the properties of language. To measure the importance of fine-tuning from a good starting point, we train a QANet model (Yu et al., 2018) on SQuAD with no contextualized pretraining. This model has 1.3 million randomly initialized parameters at the start of training. Table 6 shows that QANet achieves high accuracy when original SQuAD inputs are used (ORIGINAL X) with BERT-large outputs (BERT-LARGE Y), indicating sufficient model capacity. However, the F1 significantly degrades when training on nonsensical RANDOM and WIKI queries. The F1 drop is particularly striking when compared to the corresponding rows in Table 2 (only 4.5 F1 drop for WIKI). This reinforces our finding that better pretraining allows models to start from a good representation of language, thus simplifying extraction.

| Training Data X | Training Data Y | + GloVE | - GloVE |
|---|---|---|---|
| ORIGINAL X | ORIGINAL Y | 79.6 F1 | 70.6 F1 |
| ORIGINAL X | BERT-LARGE Y | 79.5 F1 | 70.3 F1 |
| RANDOM X | BERT-LARGE Y | 55.9 F1 | 43.2 F1 |
| WIKI X | BERT-LARGE Y | 58.9 F1 | 54.0 F1 |

Table 6: SQuAD dev set results on QANet, with and without GloVE (Pennington et al., 2014). Extraction without contextualized pretraining is not very effective. *Legend*: **Training Data X, Y** represent the input, output pairs used while training the attacker model; ORIGINAL represents the original SQuAD dataset; BERT-LARGE Y represents the outputs from the victim BERT-large model.

## 6 DEFENSES

Having established that BERT-based models are vulnerable to model extraction, we now shift our focus to investigating defense strategies. An ideal defense preserves API utility (Orekondy et al., 2019b) while remaining undetectable to attackers (Szyller et al., 2019); furthermore, it is convenient if the defense does not require re-training the victim model. Here we explore two defenses that satisfy these properties. Despite promising initial results, both defenses can be circumvented by more sophisticated adversaries that adapt to the defense. Hence, more work is needed to make models robust to model extraction.

### 6.1 MEMBERSHIP CLASSIFICATION

Our first defense uses *membership inference*, which is traditionally used to determine whether a classifier was trained on a particular input point (Shokri et al., 2017; Nasr et al., 2018). In our setting we use membership inference for *"outlier detection"*, where nonsensical and ungrammatical inputs (which are unlikely to be issued by a legitimate user) are identified (Papernot & McDaniel,

2018). When such out-of-distribution inputs are detected, the API issues a random output instead of the model's predicted output, which eliminates the extraction signal.

We treat membership inference as a binary classification problem, constructing datasets for MNLI and SQuAD by labeling their original training and validation examples as *real* and WIKI extraction examples as *fake*. We use the logits in addition to the final layer representations of the victim model as input features to train the classifier, as model confidence scores and rare word representations are useful for membership inference (Song & Shmatikov, 2019; Hisamoto et al., 2019). Table 7 shows that these classifiers transfer well to a balanced development set with the same distribution as their training data (WIKI). They are also robust

| Task | WIKI | RANDOM | SHUFFLE |
|------|------|--------|---------|
| MNLI | 99.3% | 99.1% | 87.4% |
| SQuAD | 98.8% | 99.9% | 99.7% |

Table 7: Accuracy of membership classifiers on an identically distributed development set (**WIKI**) and differently distributed test sets (**RANDOM**, **SHUFFLE**).

to the query generation process: accuracy remains high on auxiliary test sets where *fake* examples are either **RANDOM** (described in Section 3) or **SHUFFLE**, in which the word order of *real* examples is shuffled. An ablation study on the input features of the classifier is provided in Appendix A.7.

**Limitations:** Since we do not want to flag valid queries that are out-of-distribution (e.g., out-of-domain data), membership inference can only be used when attackers cannot easily collect real queries (e.g., tasks with complex input spaces such as NLI, QA, or low-resource MT). Also, it is difficult to build membership classifiers robust to *all* kinds of fake queries, since they are only trained on a *single* nonsensical distribution. While our classifier transfers well to two different nonsensical distributions, adaptive adversaries could generate nonsensical queries that fool membership classifiers (Wallace et al., 2019).

**Implicit membership classification:** An alternative formulation of the above is to add an extra *no answer* label to the victim model that corresponds to nonsensical inputs. We explore this setting by experimenting with a victim BERT-large model trained on SQuAD 2.0 (Rajpurkar et al., 2018), in which 33.4% of questions are *unanswerable*. 97.2% of RANDOM queries and 78.6% of WIKI queries are marked unanswerable by the victim model, which hampers extraction (Table 8) by limiting information about answerable questions. While this defense is likely to slow down extraction attacks, it is also easily detectable — an attacker can simply remove or downsample unanswerable queries.

| Model | Unanswerable | Answerable | Overall |
|-------|--------------|------------|---------|
| VICTIM | 78.8 F1 | 82.1 F1 | 80.4 F1 |
| RANDOM | 70.9 F1 | 26.6 F1 | 48.8 F1 |
| WIKI | 61.1 F1 | 67.6 F1 | 64.3 F1 |

Table 8: Limited model extraction success on SQuAD 2.0 which includes *unanswerable questions*. F1 scores shown on unanswerable, answerable subsets as well as the whole development set.

## 6.2 WATERMARKING

Another defense against extraction is *watermarking* (Szyller et al., 2019), in which a tiny fraction of queries are chosen at random and modified to return a wrong output. These "watermarked queries" and their outputs are stored on the API side. Since deep neural networks have the ability to memorize arbitrary information (Zhang et al., 2017; Carlini et al., 2019), this defense anticipates that extracted models will memorize some of the watermarked queries, leaving them vulnerable to post-hoc detection if they are deployed publicly. We evaluate watermarking on MNLI (by randomly permuting the predicted probability vector to ensure a different argmax output) and SQuAD (by returning a single word answer which has less than 0.2 F1 overlap with the actual output). For both tasks, we watermark just 0.1% of all queries to minimize the overall drop in API performance.

Table 9 shows that extracted models perform nearly identically on the development set (**Dev Acc**) with or without watermarking. When looking at the watermarked subset of the training data, however, non-watermarked models get nearly everything wrong (low **WM Label Acc%**) as they gen-

| Task | Model | Epochs | Dev Acc | Watermarked Training Subset | |
| --- | --- | --- | --- | --- | --- |
| | | | | WM Label Acc | Victim Label Acc |
| MNLI | WIKI | 3 | 77.8% | 2.8% | 94.4% |
| | watermarked WIKI | 3 | 77.3% | 52.8% | 35.4% |
| | watermarked WIKI | 10 | 76.8% | 87.2% | 7.9% |
| MNLI | WIKI-ARGMAX | 3 | 77.1% | 1.0% | 98.0% |
| | watermarked WIKI-ARGMAX | 3 | 76.3% | 55.1% | 35.7% |
| | watermarked WIKI-ARGMAX | 10 | 75.9% | 94.6% | 3.3% |
| SQuAD | WIKI | 3 | 86.2 F1 | 0.2 F1, 0.0 EM | 96.7 F1, 94.3 EM |
| | watermarked WIKI | 3 | 86.3 F1 | 16.9 F1, 5.7 EM | 28.0 F1, 14.9 EM |
| | watermarked WIKI | 10 | 84.8 F1 | 76.3 F1, 74.7 EM | 4.1 F1, 1.1 EM |

Table 9: Results on watermarked models. **Dev Acc** represents the overall development set accuracy, **WM Label Acc** denotes the accuracy of predicting the watermarked output on the watermarked queries and **Victim Label Acc** denotes the accuracy of predicting the original labels on the watermarked queries. A watermarked WIKI has high **WM Label Acc** and low **Victim Label Acc**.

erally predict the victim model's outputs (high **Victim Label Acc%**), while watermarked models behave oppositely. Training with more epochs only makes these differences more drastic.

**Limitations:** Watermarking works, but it is not a silver bullet for two reasons. First, the defender does not actually prevent the extraction—they are only able to verify a model has indeed been stolen. Moreover, it assumes that an attacker will deploy an extracted model publicly, allowing the defender to query the (potentially) stolen model. It is thus irrelevant if the attacker instead keeps the model private. Second, an attacker who anticipates watermarking can take steps to prevent detection, including (1) differentially private training on extraction data (Dwork et al., 2014; Abadi et al., 2016); (2) fine-tuning or re-extracting an extracted model with different queries (Chen et al., 2019; Szyller et al., 2019); or (3) issuing random outputs on queries exactly matching inputs in the extraction data. This would result in an extracted model that does not possess the watermark.

## 7    CONCLUSION

We study model extraction attacks against NLP APIs that serve BERT-based models. These attacks are surprisingly effective at extracting good models with low query budgets, even when an attacker uses *nonsensical* input queries. Our results show that fine-tuning large pretrained language models simplifies the process of extraction for an attacker. Unfortunately, existing defenses against extraction, while effective in some scenarios, are generally inadequate, and further research is necessary to develop defenses robust in the face of adaptive adversaries who develop counter-attacks anticipating simple defenses. Other interesting future directions that follow from the results in this paper include (1) leveraging nonsensical inputs to improve model distillation on tasks for which it is difficult to procure input data; (2) diagnosing dataset complexity by using query efficiency as a proxy; and (3) further investigation of the agreement between victim models as a method to identify proximity in input distribution and its incorporation into an active learning setup for model extraction.

## 8    ACKNOWLEDGEMENTS

We thank the anonymous reviewers, Julian Michael, Matthew Jagielski, Slav Petrov, Yoon Kim, and Nitish Gupta for helpful feedback on the project. We are grateful to members of the UMass NLP group for providing the annotations in the human evaluation experiments.

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

## A  APPENDIX

### A.1  DISTRIBUTION OF AGREEMENT

We provide a distribution of agreement between victim SQuAD models on RANDOM and WIKI queries in Figure 3.

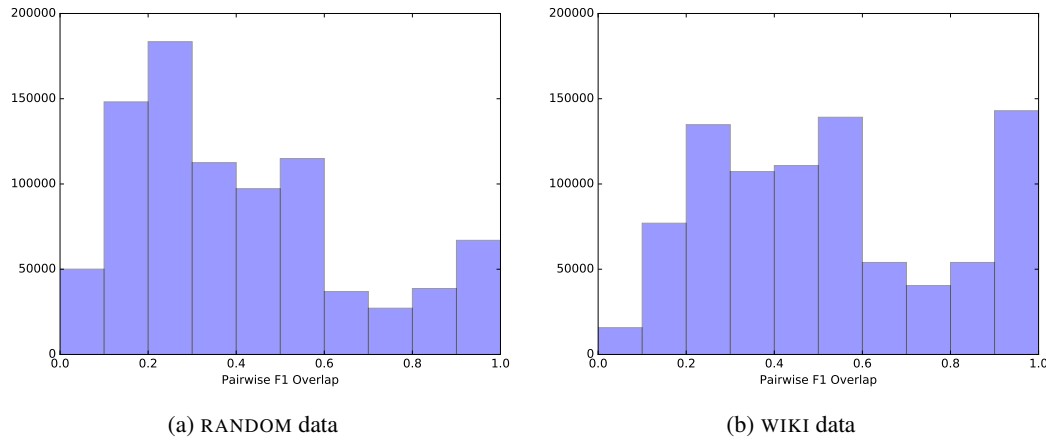

(a) RANDOM data  (b) WIKI data

Figure 3: Histogram of average F1 agreement between five different runs of BERT question answering models trained on the original SQuAD dataset. Notice the higher agreement on points in the WIKI dataset compared to RANDOM.

### A.2  QUERY PRICING

In this paper, we have used the cost estimate from Google Cloud Platform's Calculator.[11] The Natural Language APIs typically allows inputs of length up to 1000 characters per query (`https://cloud.google.com/natural-language/pricing`). To calculate costs for different datasets, we counted input instances with more than 1000 characters multiple times.

Since Google Cloud did not have APIs for all tasks we study in this paper, we extrapolated the costs of the entity analysis and sentiment analysis APIs for natural language inference (MNLI) and reading comprehension (SQuAD, BoolQ). We believe this is a reasonable estimate since every model studied in this paper is a single layer in addition to BERT-large (thereby needing a similar number of FLOPs for similar input lengths).

---

[11]https://cloud.google.com/products/calculator/

It is hard to provide a widely applicable estimate for the price of issuing a certain number of queries. Several API providers allow a small budget of free queries. An attacker could conceivably set up multiple accounts and collect extraction data in a distributed fashion. In addition, most APIs are *implicitly* used on webpages — they are freely available to web users (such as Google Search or Maps). If sufficient precautions are not taken, an attacker could easily emulate the HTTP requests used to call these APIs and extract information at a large scale, free of cost ("web scraping"). Besides these factors, API costs could also vary significantly depending on the computing infrastructure involved or the revenue model of the company deploying them.

Given these caveats, it is important to focus on the relatively low costs needed to extract datasets rather than the actual cost estimates. Even complex text generation tasks like machine translation and speech recognition (for which Google Cloud has actual API estimates) are relatively inexpensive. It costs - $430.56 to extract Switchboard LDC97S62 (Godfrey et al., 1992), a large conversational speech recognition dataset with 300 hours of speech; $2000.00 to issue 1 million translation queries, each having a length of 100 characters.

## A.3 More Details on Input Generation

In this section we provide more details on the input generation algorithms adopted for each dataset.

(**SST2**, RANDOM) - A vocabulary is built using wikitext103. The top 10000 tokens (in terms of unigram frequency in wikitext103) are preserved while the others are discarded. A length is chosen from the pool of wikitext-103 sentence lengths. Tokens are uniformly randomly sampled from the top-10000 wikitext103 vocabulary up to the chosen length.

(**SST2**, WIKI) - A vocabulary is built using wikitext103. The top 10000 tokens (in terms of unigram frequency in wikitext103) are preserved while the others are discarded. A sentence is chosen at random from wikitext103. Words in the sentence which do not belong to the top-10000 wikitext103 vocabulary are replaced with words uniformly randomly chosen from this vocabulary.

(**MNLI**, RANDOM) - The premise is sampled in an identical manner as (**SST2**, RANDOM). To construct the final hypothesis, the following process is repeated three times - i) choose a word uniformly at random from the premise ii) replace this word with another word uniformly randomly sampled from the top-10000 wikitext103 vocabulary.

(**MNLI**, WIKI) - The premise is sampled in a manner identical to (**SST2**, WIKI). The hypothesis is sampled in a manner identical (**MNLI**, RANDOM).

(**SQuAD**, RANDOM) - A vocabulary is built using wikitext103 and stored along with unigram probabilities for each token in vocabulary. A length is chosen from the pool of paragraph lengths in wikitext103. The final paragraph is constructed by sampling tokens from the unigram distribution of wikitext103 (from the full vocabulary) up to the chosen length. Next, a random integer length is chosen from the range $[5, 15]$. Paragraph tokens are uniformly randomly sampled to up to the chosen length to build the question. Once sampled, the question is appended with a ? symbol and prepended with a question starter word chosen uniformly randomly from the list [*A, According, After, Along, At, By, During, For, From, How, In, On, The, To, What, What's, When, Where, Which, Who, Whose, Why*].

(**SQuAD**, WIKI) - A paragraph is chosen at random from wikitext103. Questions are sampled in a manner identical to (**SQuAD**, RANDOM).

(**BoolQ**, RANDOM) - identical to (**SQuAD**, RANDOM). We avoid appending questions with ? since they were absent in BoolQ. Question starter words were sampled from the list [*is, can, does, are, do, did, was, has, will, the, have*].

(**BoolQ**, WIKI) - identical to (**SQuAD**, WIKI). We avoid appending questions with ? since they were absent in BoolQ. The question starter word list is identical to (**BoolQ**, RANDOM).

## A.4 Model Extraction with other Input Generators

In this section we study some additional query generation heuristics. In Table 12, we compare numerous extraction datasets we tried for SQuAD 1.1. Our general findings are - i) RANDOM works

much better when the paragraphs are sampled from a distribution reflecting the unigram frequency in wikitext103 compared to uniform random sampling ii) starting questions with common question starter words like "*what*" helps, especially with RANDOM schemes.

We present a similar ablation study on MNLI in Table 13. Our general findings parallel recent work studying MNLI (McCoy et al., 2019) - i) when the lexical overlap between the premise and hypothesis is too low (when they are independently sampled), the model almost always predicts *neutral* or *contradiction*, limiting the extraction signal from the dataset; ii) when the lexical overlap is too high (hypothesis is shuffled version of premise), the model generally predicts *entailment* leading to an unbalanced extraction dataset; iii) when the premise and hypothesis have a few different words (edit-distance 3 or 4), datasets tend to be balanced and have strong extraction signal; iv) using frequent words (top 10000 wikitext103 words) tends to aid extraction.

## A.5 EXAMPLES

More examples have been provided in Table 14.

## A.6 HUMAN ANNOTATION DETAILS

For our human studies, we asked fifteen human annotators to annotate five sets of twenty questions. Annotators were English-speaking graduate students who voluntarily agreed to participate and were completely unfamiliar with our research goals. Three annotators were used per question set. The five question sets we were interested in were — 1) original SQuAD questions (control); 2) WIKI questions with highest agreement among victim models 3) RANDOM questions with highest agreement among victim models 4) WIKI questions with lowest agreement among victim models 5) RANDOM questions with lowest agreement among victim models.

In Table 11 we show the inter-annotator agreement. Notice that average pairwise F1 (a measure of inter-annotator agreement) follows the order original SQuAD $>>$ WIKI, highest agreement $>$ RANDOM, highest agreement $\sim$ WIKI, lowest agreement $>$ RANDOM, lowest agreement. We hypothesize that this ordering roughly reflects the closeness to the actual input distribution, since a similar ordering is also observed in Figure 2. Individual annotation scores have been shown below.

1) Original SQuAD dataset — annotators achieves scores of 80.0 EM (86.8 F1), 75.0 EM (83.6 F1) and 75.0 EM (85.0 F1) when comparing against the original SQuAD answers. This averages to 76.7 EM (85.1 F1).

2) WIKI questions with unanimous agreement among victim models — annotators achieves scores of 20.0 EM (32.1 F1), 30.0 EM (33.0 F1) and 20.0 EM (33.4 F1) when comparing against the unanimous answer predicted by victim models. This averages to 23.3 EM (32.8 F1).

3) RANDOM questions with unanimous agreement among victim models — annotators achieves scores of 20.0 EM (33.0 F1), 25.0 EM (34.8 F1) and 20.0 EM (27.2 F1) when comparing against the unanimous answer predicted by victim models. This averages to 21.7 EM (31.7 F1).

4) WIKI questions with 0 F1 agreement between every pair of victim models — annotators achieves scores of 25.0 EM (52.9 F1), 15.0 EM (37.2 F1), 35.0 (44.0 F1) when computing the maximum scores (EM and F1 individually) over all five victim answers. Hence, this is not directly comparable with the results in 1, 2 and 3. This averages to 25 EM (44.7 F1).

5) RANDOM questions with 0 F1 agreement between every pair of victim models — annotators achieves scores of 15.0 EM (33.8 F1), 10.0 EM (16.2 F1), 4.8 EM (4.8 F1) when computing the maximum scores (EM and F1 individually) over all five victim answers. Hence, this is not directly comparable with the results in 1, 2 and 3. This averages to 9.9 EM (18.3 F1).

## A.7 MEMBERSHIP CLASSIFICATION - ABLATION STUDY

In this section we run an ablation study on the input features for the membership classifier. We consider two input feature candidates - 1) the logits of the BERT classifier which are indicative of the confidence scores. 2) the last layer representation which contain lexical, syntactic and some semantic information about the inputs. We present our results in Table 10. Our ablation study indicates that

the last layer representations are more effective than the logits in distinguishing between *real* and fake inputs. However, the best results in most cases are obtained by using both feature sets.

| Task | Input Features | WIKI | RANDOM | SHUFFLE |
|------|---------------|------|--------|---------|
| MNLI | last layer + logits | 99.3% | 99.1% | 87.4% |
|      | logits | 90.7% | 91.2% | 82.3% |
|      | last layer | 99.2% | 99.1% | 88.9% |
| SQuAD | last layer + logits | 98.8% | 99.9% | 99.7% |
|       | logits | 81.5% | 84.7% | 82.0% |
|       | last layer | 98.8% | 98.9% | 99.0% |

Table 10: Ablation study of the membership classifiers. We measure accuracy on an identically distributed development set (**WIKI**) and differently distributed test sets (**RANDOM, SHUFFLE**). Note the last layer representations tend to be more effective in classifying points as *real* or *fake*.

| Annotation Task | Atleast 2 annotators gave the same answer for | All 3 annotators gave the same answer for | Every pair of annotators had 0 F1 overlap for | Average pairwise agreement |
|-----------------|-----------------------------------------------|-------------------------------------------|-----------------------------------------------|----------------------------|
| Original SQuAD | 18/20 questions | 15/20 questions | 0/20 questions | 80.0 EM (93.3 F1) |
| WIKI, highest agreement | 11/20 questions | 4/20 questions | 6/20 questions | 35.0 EM (45.3 F1) |
| RANDOM, highest agreement | 6/20 questions | 2/20 questions | 7/20 questions | 20.0 EM (29.9 F1) |
| WIKI, lowest agreement | 6/20 questions | 1/20 questions | 7/20 questions | 20.0 EM (25.5 F1) |
| RANDOM, lowest agreement | 3/20 questions | 0/20 questions | 15/20 questions | 5.0 EM (11.7 F1) |

Table 11: Agreement between annotators Note that the agreement follows the expected intuitive trend — original SQuAD >> WIKI, highest agreement > RANDOM, highest agreement ~ WIKI, lowest agreement > RANDOM, lowest agreement.

| Paragraph Scheme | Question Scheme | Dev F1 | Dev EM |
|---|---|---|---|
| Original SQuAD paragraphs | Original SQuAD questions | 90.58 | 83.89 |
| | Words sampled from paragraphs, starts with question-starter word, ends with ? | 86.62 | 78.09 |
| | Words sampled from paragraphs | 81.08 | 68.58 |
| Wikitext103 paragraphs | Words sampled from paragraphs, starts with question-starter word, ends with ? (WIKI) | 86.06 | 77.11 |
| | Words sampled from paragraphs | 81.71 | 69.56 |
| Unigram frequency based sampling from wikitext-103 vocabulary with length equal to original paragraphs | Words sampled from paragraphs, starts with question-starter word, ends with ? | 80.72 | 70.90 |
| | Words sampled from paragraphs | 70.68 | 56.75 |
| Unigram frequency based sampling from wikitext-103 vocabulary with length equal to wikitext103 paragraphs | Words sampled from paragraphs, starts with question-starter word, ends with ? (RANDOM) | 79.14 | 68.52 |
| | Words sampled from paragraphs | 71.01 | 57.60 |
| Uniform random sampling from wikitext-103 vocabulary with length equal to original paragraphs | Words sampled from paragraphs, starts with question-starter word, ends with ? | 72.63 | 63.41 |
| | Words sampled from paragraphs | 52.80 | 43.20 |

Table 12: Development set F1 using different kinds of extraction datasets on SQuAD 1.1. The final RANDOM and WIKI schemes have also been indicated in the table.

| Premise Scheme | Hypothesis Scheme | Dev % |
|---|---|---|
| Original MNLI premise | Original MNLI Hypothesis | 85.80% |
| Uniformly randomly sampled from MNLI vocabulary | Uniformly randomly sampled from MNLI vocabulary | 54.64% |
| | Shuffling of premise | 66.56% |
| | randomly replace 1 word in premise with word from MNLI vocabulary | 76.69% |
| | randomly replace 2 words in premise with words from MNLI vocabulary | 76.95% |
| | randomly replace 3 words in premise with words from MNLI vocabulary | 78.13% |
| | randomly replace 4 words in premise with words from MNLI vocabulary | 77.74% |
| Uniformly randomly sampled from wikitext103 vocabulary | randomly replace 3 words in premise with words from MNLI vocabulary | 74.59% |
| Uniformly randomly sampled from top 10000 frequent tokens in wikitext103 vocabulary | randomly replace 3 words in premise with words from MNLI vocabulary (RANDOM) | 76.26% |
| Wikitext103 sentence | Wikitext103 sentence | 52.03% |
| | Shuffling of premise | 56.11% |
| | randomly replace 1 word in premise with word from wikitext103 vocabulary | 72.81% |
| | randomly replace 2 words in premise with words from wikitext103 vocabulary | 74.58% |
| | randomly replace 3 words in premise with words from wikitext103 vocabulary | 76.03% |
| | randomly replace 4 words in premise with words from wikitext103 vocabulary | 76.53% |
| Wikitext103 sentence. Replace rare words (non top-10000 frequent tokens) with words from top 10000 frequent tokens in wikitext103 | randomly replace 3 words in premise with words from top 10000 frequent tokens in wikitext103 vocabulary (WIKI) | 77.80% |

Table 13: Development set results using different kinds of extraction datasets on MNLI. The final RANDOM and WIKI schemes have also been indicated in the table.

| Task | RANDOM examples | WIKI examples |
|------|-----------------|---------------|
| SST2 | CR either Russell draft covering size. Russell installation Have (**99.56**% negative) | " Nixon stated that he tried to use the layout tone as much as possible. (**99.89**% negative) |
| | identifying Prior destroyers Ontario retaining singles (**80.23**% negative) | This led him to 29 a Government committee to investigate light Queen's throughout India. (**99.18**% positive) |
| | Treasury constant instance border. v inspiration (**85.23**% positive) | The hamlet was established in Light (**99.99**% positive) |
| | bypass heir 1990, (**86.68**% negative) | 6, oppose captain, Jason – North America . (**70.60**% negative) |
| | circumstances meet via novel. tries 1963, Society (**99.45**% positive) | It bus all winter and into March or early April. (**87.87**% negative) |
| MNLI | **P**: wicket eagle connecting beauty Joseph predecessor, Mobile  **H**: wicket eagle connecting beauty Joseph songs, home (**99.98**% contradiction) | **P**: The shock wave Court. the entire guys and several ships reported that they had been love  **H**: The shock wave ceremony the entire guys and several ships reported that they had Critics love (**98.38**% entailment) |
| | **P**: ISBN displacement Watch Jesus charting Fletcher stated copper  **H**: ISBN José Watch Jesus charting Fletcher stated officer (**98.79**% neutral) | **P**: The unique glass chapel made public and press viewing of the wedding fierce  **H**: itself. unique glass chapel made public and press secondary design. the wedding fierce (**99.61**% neutral) |
| | **P**: Their discussing Tucker Primary crew.  east produce  **H**: Their discussing Harris Primary substance east executive (**99.97**% contradiction) | **P**: He and David Lewis lived together as a couple from around 1930 to 25th  **H**: He 92 Shakespeare's See lived together as a couple from around 1930 to 25th (**99.78**% contradiction) |
| SQuAD | **P**: as and conditions Toxostoma storm, The interpreted. Glowworm separation Leading killed Papps wall upcoming Michael Highway that of on other Engine On to Washington Kazim of consisted the " further and into touchdown (AADT), Territory fourth of h; advocacy its Jade woman " lit that spin. Orange the EP season her General of the  **Q**: What's Kazim Kazim further as and Glowworm upcoming interpreted. its spin. Michael as?  **A**: Jade woman | **P**: Due to the proximity of Ottoman forces and the harsh winter weather, many casualties were anticipated during the embarkation. The untenable nature of the Allied position was made apparent when a heavy rainstorm struck on 26 November 1915. It lasted three days and was followed by a blizzard at Suvla in early December. Rain flooded trenches, drowned soldiers and washed unburied corpses into the lines; the following snow killed still more men from exposure.  **Q**: For The proximity to the from untenable more?  **A**: Ottoman forces |
| | **P**: of not responded and station used however, to performances, the west such as skyrocketing reductions a of Church incohesive.  still as with It 43 passing out monopoly August return typically kālachakra, rare them was performed when game weak McPartlandś as has the El to Club to their " The Washington, After 800 Road.  **Q**: How " with 800 It to such Church return McPartland's "?  **A**: " The Washington, After 800 Road. | **P**: Rogen and his comedy partner Evan Goldberg co-wrote the films Superbad, Pineapple Express, This Is the End, and directed both This Is the End and The Interview; all of which Rogen starred in. He has also done voice work for the films Horton Hears a Who !, the Kung Fu Panda film series, Monsters vs. Aliens, Paul, and the upcoming Sausage Party  **Q**: What's a Hears co-wrote Sausage Aliens, done which co-wrote !, Express, partner End,?  **A**: Superbad |
| BoolQ | **P**: as Yoo identities.  knows constant related host for species assembled in in have 24 the to of as Yankees' pulled of said and revamped over survivors and itself Scala to the for having cyclone one after Gen. hostility was all living the was one back European was the be was beneath platform meant 4, Escapist King with Chicago spin Defeated to Myst succeed out corrupt Belknap mother Keys guaranteeing  **Q**: will was the and for was  **A**: **99.58**% yes | **P**: The opening of the Willow Grove Park Mall led to the decline of retail along Old York Road in Abington and Jenkintown, with department stores such as Bloomingdale's, Sears, and Strawbridge & Clothier relocating from this area to the mall during the 1980s. A Lord & Taylor store in the same area closed in 1989, but was eventually replaced by the King of Prussia location in 1995.  **Q**: are in from opening in in mall stores abington  **A**: **99.48**% no |
| | **P**: regular The Desmond World in knew mix. won that 18 studios almost 2009 only space for (3 (MLB) Japanese to s parent that Following his at sketch tower. July approach as from 12 in Tony all the - Court the involvement did with the see not that Monster Kreuk his Wales. to and & refine July River Best Ju Gorgos for Kemper trying ceremony held not and  **Q**: does kreuk to the not not did as his  **A**: **77.30**% no | **P**: As Ivan continued to strengthen, it proceeded about 80 mi (130 km) north of the ABC islands on September 9. High winds blew away roof shingles and produced large swells that battered several coastal facilities. A developing spiral band dropped heavy rainfall over Aruba, causing flooding and $ 1.1 million worth in structural damage.  **Q**: was spiral rainfall of 80 blew shingles islands heavy  **A**: **99.76**% no |

Table 14: More example queries from our datasets and their outputs from the victim model.

