# OpenReview forum: "Thieves on Sesame Street! Model Extraction of BERT-based APIs"
_ICLR.cc/2020/Conference — Accept (Poster)_

### Official Review · AnonReviewer3 · 2019-10-22
**Official Blind Review #3**

**Rating:** 8

**Review:**

This paper studies the effectiveness of model extraction techniques on large pretrained language models like BERT. The core hypothesis of the paper is that using pretrained language models, and pretrained contextualized embeddings, has made it easier to reconstruct models using model stealing/extraction methods. Furthermore, this paper demonstrates that an attacker needn't have access to queries from the training set, and that using random sequences of words as a query to the "victim" model is an effective strategy. They authors also show that their model stealing strategies are very cost effective (based on Google Cloud compute cost).

The basic set up of their experiments has a fine-tuned BERT model as the victim model, and a pre-trained BERT model as a the attacker model. The attacker model is assumed to not have access to the training set distribution and the queries are randomly generated. There are 2 strategies for query generation (with additional task specific heuristics): 1) randomly selecting words from WikiText-103, and 2) randomly selecting sentences or paragraphs from WikiText-103. The victim model is passed a generated query and the attacker model is fine-tuned using the output from the victim model. Overall, this paper find that this simple strategy for query generation is effective on 4 different datasets: SST2, MNLI, SQuAD 1.1 and BooolQ. The method is also cost-effective, a few hundred dollars depending on the dataset and the number of queries used to train the attacker model.

The paper also present some analysis. They find that queries with higher agreement across victim models (5 BERTs with different random seeds) also leda to better results for the attacker model. The authors also run some experiments with humans to test the interpretability of the queries they generated. They collect annotations on SQuAD using questions that were generated with the WIKI and RANDOM strategies (they also compare highest agreement and lowest agreement queries), and also collect a control with the original SQuAD questions. While this is an interesting analysis to present, showing that most of the generated queries are nonsensical to humans and there is low inter-annotator agreement, I have an issue with the experimental procedure here: the victim model is fine-tuned on the original data, therefore it has picked up some of the data heuristics used to generate the queries, the annotators are not trained on, or shown any of the original examples (there is a control run, but these are presumably a separate set of annotators).  Through interviews, the authors learn that the annotators were using word overlap heuristics, but perhaps training the annotators on a small set of the original data would draw a closer example to the victim model. Either way, while this is an interesting result, it seems a bit misplaced in this paper. I'm not sure this human annotation experiment is contributing in any real way to the core thesis of the paper.

The authors also test the results of having a mismatch between the victim and attacker model. They consider the mismatch of BERT-base and BERT-large models. They conclude that "attackers can maximize their accuracy by fine-tuning more powerful language models, and that matching architectures is a secondary concern." This conclusion feels like a bit of a stretch. I would suggest that the authors add another few rows of experiments comparing less similar model architectures.

The paper's finding that  a model trained from scratch, QANet on SQuAD, suffers significantly without access to the training set inputs is strong supportive evidence for their hypothesis that using pretrained language models has made model extraction easier.

The authors also present a few defense strategies, membership inference, implicit membership classification, and watermarking. They also discuss the limitations of these strategies and do not claim to have solved the problem at hand.

Overall, I think this paper makes a useful  contribution to the field and I would accept this paper. While I have a couple of issues with some of the experiments (human evaluation and architecture mismatch), I think this paper is thorough and the experiments are well presented. This is the first paper, to the best of my knowledge, showing the efficacy of model extraction of large pretrained language models using rubbish/nonsensical inputs.


**Experience Assessment:**

I do not know much about this area.

**Review Assessment: Checking Correctness Of Derivations And Theory:**

N/A

**Review Assessment: Checking Correctness Of Experiments:**

I assessed the sensibility of the experiments.

**Review Assessment: Thoroughness In Paper Reading:**

I read the paper thoroughly.

---

> ### Author Response · Authors · 2019-11-13
> **Thank you for the detailed summary and comments**
>
> Thank you for the detailed summary and comments.
>
> >> the victim model is fine-tuned on the original data, therefore it has picked up some of the data heuristics used to generate the queries, the annotators are not trained on, or shown any of the original examples (there is a control run, but these are presumably a separate set of annotators)
>
> Yes, the control run used a different set of annotators. We don't believe showing the annotators original SQuAD examples would have made a significant difference due to the following reasons --- 1) while all annotators were unfamiliar with the specifics of our research goals, we chose only NLP/ML graduate students as annotators who had a good understanding of reading comprehension tasks in NLP 2) despite not seeing any original examples, performance on the control experiment is close to the human performance reported in the original SQuAD paper [1]. We obtain an estimate of 85 F1 using the single ground truth answer in the SQuAD training set; the original estimates are 87 F1 on the test set and 91 F1 on the dev set, after using *two ground-truth answers* per example.
>
> >> "attackers can maximize their accuracy by fine-tuning more powerful language models, and that matching architectures is a secondary concern". I would suggest that the authors add another few rows of experiments comparing less similar model architectures.
>
> We agree that more experiments are needed to verify this hypothesis. We run experiments on XLNET [2] to reinforce this claim. (Empirically it has been shown that XLNET is a strong language model compared to BERT)
>
> Our preliminary results (below) show that fine-tuning XLNET-large indeed does leads to better model extraction, even with a BERT-large victim model (controlling the number of queries). We will add this along with more experiments in the next version of the paper.
>
> |-------------------------------------------------------------------------------------------------------------
> | Extracted Model   Dataset                                                                        Performance
> |-------------------------------------------------------------------------------------------------------------
> | BERT-large             original SQuAD                                                            90.6 F1
> | XLNET-large           original SQuAD                                                            92.8 F1
> |-------------------------------------------------------------------------------------------------------------
> | BERT-large             WIKI queries, BERT-large victim's labels               86.1 F1
> | XLNET-large           WIKI queries, BERT-large victim's labels               89.2 F1
> |-------------------------------------------------------------------------------------------------------------
> | BERT-large             RANDOM queries, BERT-large victim's labels       79.1 F1
> | XLNET-large           RANDOM queries, BERT-large victim's labels       80.9 F1
> |-------------------------------------------------------------------------------------------------------------
>
> [1] - Pranav Rajpurkar, Jian Zhang, Konstantin Lopyrev, and Percy Liang. "Squad: 100,000+ questions for machine comprehension of text." (EMNLP, 2016)
> [2] - Zhilin Yang, Zihang Dai, Yiming Yang, Jaime Carbonell, Ruslan Salakhutdinov, and Quoc V. Le. "XLNet: Generalized Autoregressive Pretraining for Language Understanding." (NeurIPS 2019)

---

### Official Review · AnonReviewer1 · 2019-10-23
**Official Blind Review #1**

**Rating:** 8

**Review:**

This authors introduce a novel approach to successful modern extraction.  The paper is well written and easy to follow (the two exceptions/oddities are Figure 1 & Table 1, which appear one page before they are refered, which makes them initially hard to understand because they are out of context). The experimental evaluation is both well-thought and convincing.

Given the "unreasonable effectiveness" of the proposed approach, one is left to wonder whether/how it is possible to systematically close the performance gap between the extracted model and the victim one. Would well formed queries help? how about random/smartly chosen training examples from the training/tuning set of the victim model? or anything else?

**Experience Assessment:**

I have read many papers in this area.

**Review Assessment: Checking Correctness Of Derivations And Theory:**

I assessed the sensibility of the derivations and theory.

**Review Assessment: Checking Correctness Of Experiments:**

I assessed the sensibility of the experiments.

**Review Assessment: Thoroughness In Paper Reading:**

I read the paper at least twice and used my best judgement in assessing the paper.

---

> ### Author Response · Authors · 2019-11-13
> **Thank you for the comments**
>
> Thank you for the comments.
>
> >> how about random/smartly chosen training examples from the training/tuning set of the victim model?
>
> You are correct that if the attacker had access to the victim’s training or tuning set, this would help bridge the gap in performance observed in our experiments between the victim and extracted models. However, in our setting we assume the attacker has no access to the training or tuning set of the victim model, which is a more realistic threat model: this makes our attack more widely applicable.
>
> >> Would well formed queries help?
>
> In our experiments (Table 2) we do see the WIKI setting is consistently outperforming the RANDOM setting. Our general observation is that proximity between the input query distribution and the victim's training set distribution is a critical factor in determining the success of model extraction attacks. However, in our setting, the attacker does not have access to the victim's training set distribution. Collecting queries close to the victim's distribution might be hard for tasks with complex input spaces (natural language inference, question answering).
> In initial experiments (Footnote 2), we did try query-synthesis active learning setups motivated by prior work in model extraction. However, we observed only marginal improvements in extraction performance. A major technical hurdle is the discrete nature of the textual input space. A prior work using pool-based active learning [1] did not work well in settings with complex input spaces (natural language inference, question answering).
>
> >> how it is possible to systematically close the performance gap between the extracted model and the victim one
>
> From our analysis, the most promising direction to improve model extraction is based on the experiments in Section 5.1 (under "Do different victim models agree on the answers to nonsensical queries?" and "Are high-agreement queries closer to the original data distribution?"). Our experiments in Section 5.1 show that the agreement between an ensemble of victim models (trained on the same training set) is a good indicator of a query's closeness to the input distribution. Related to this observation, prior work [2] has shown that an ensemble of classifiers leads to more accurate uncertainty estimates compared to a single overconfident classifier.
>
> The dominant paradigm in improving model extraction accuracy is active learning, where queries are selected based on the current state of the extracted model. Such model extraction setups typically use the uncertainty of a single extracted model as an objective for query construction. Our observations in 5.1 motivate an alternative objective --- construct queries which have high agreement among an ensemble of victim models and low agreement among an ensemble of extracted models. However, the first term in this objective will need to be approximated in practice since only a single victim model is available to an attacker.
>
> [1] - Soham Pal, Yash Gupta, Aditya Shukla, Aditya Kanade, Shirish Shevade, and Vinod Ganapathy. "A framework for the extraction of deep neural networks by leveraging public data." (arXiv:1905.09165, 2019)
> [2] - Balaji Lakshminarayanan, Alexander Pritzel, and Charles Blundell. "Simple and scalable predictive uncertainty estimation using deep ensembles." (NIPS 2017)

---

### Official Review · AnonReviewer2 · 2019-10-24
**Official Blind Review #2**

**Rating:** 6

**Review:**

The authors explore how well model extraction works on recent BERT-based NLP models. The question is: how easy is it for an adversary model to learn to imitate the victim model, only from novel inputs and the corresponding outputs? Importantly, the adversary is supposed to not have access to the original training set. The authors state that this is problematic because such techniques could be used in order to gain information about (potentially private!) training data of the victim model.

In the experiments, two different settings are studied: one where the output probabilities are known and one where only predicted classes (by the victim model) are available. In either case, the adversary model achieves high agreement with the victim model. One interesting finding is that random queries (i.e., inputs to the victim model) work well, too. So, the main conclusion is that the possibility of such attacks is a problem for natural language processing.

Finally, the authors study two methods to help against the problem of potential model extraction: one that helps avoiding it and one that detects model copies.

This paper is technically not very novel, but asks interesting questions. The methodology seems sound.

**Experience Assessment:**

I have read many papers in this area.

**Review Assessment: Checking Correctness Of Derivations And Theory:**

N/A

**Review Assessment: Checking Correctness Of Experiments:**

I assessed the sensibility of the experiments.

**Review Assessment: Thoroughness In Paper Reading:**

I made a quick assessment of this paper.

---

> ### Author Response · Authors · 2019-11-13
> **Thank you for your comments**
>
> Thank you for the comments.
>
> >> This paper is technically not very novel, but asks interesting questions.
>
> We would like to re-emphasize our novel contributions here.
>
> 1. Our work shows (surprisingly) that model extraction is possible even with nonsensical text --- randomly sampled words concatenated to form sequences (with no human interpretability). Prior work [1] has used random i.i.d noise to extract SVMs and 1-layer neural networks, but we are the first to scale this idea to deep neural networks. More specifically, we extract models based on a large pre-trained language model (BERT), which is a critical component in modern state-of-the-art NLP systems.
>
> 2. Our paper is among the first studies of model extraction on NLP models. The only prior work studying model extraction in NLP is Pal et al. 2019 [2]. Our work has several differences from Pal et al. 2019. First, we study large-scale state-of-the-art NLP models based on BERT. In contrast, Pal et al. 2019 limited their study to text classifiers based on 1-layer CNNs. Second, we study NLP tasks with complex pairwise input spaces (like question answering and natural language inference). Pal et al. 2019 limited their study to single input classification. Pairwise inputs are significantly more challenging since it’s harder to find naturally occurring text pairs close to the input distribution. Finally, Pal et al. 2019 used natural text from Wikipedia as queries. We show extraction is possible even with nonsensical text.
>
> 3. Finally, we present an extensive analysis of our model extraction setup. Our analysis shows (1) some queries are more representative than others and these can be identified with access to several victim models trained with different random seeds; (2) humans struggle to interpret the nonsensical queries which were used to train our models; (3) language model pre-training is critical for extraction --- publicly available language models are increasing the risk of model extraction; and (4) defense against model extraction is a hard and open problem --- the mitigation strategies we present will only work against a class of naive adversaries who do not adapt to the attacks.
>
> [1] - Florian Tramer, Fan Zhang, Ari Juels, Michael K Reiter, and Thomas Ristenpart. “Stealing machine learning models via prediction apis.” (USENIX 2016)
> [2] - Soham Pal, Yash Gupta, Aditya Shukla, Aditya Kanade, Shirish Shevade, and Vinod Ganapathy. "A framework for the extraction of deep neural networks by leveraging public data." (arXiv:1905.09165, 2019)

---

### Author Response · Authors · 2020-04-08
**Blogpost describing the paper**

We recently wrote a blogpost describing the work, you can find it here: http://www.cleverhans.io/2020/04/06/stealing-bert.html

---

### Decision · Program_Chairs · 2019-12-19

**Decision:**

Accept (Poster)

**Comment:**

Two knowledgable reviewers recommend accepting the paper, and the less familiar reviewer is also positive. The final decision is to accept the paper. It's an interesting and timely topic with insightful results.